# A Translational Mouse Model for NASH with Advanced Fibrosis and Atherosclerosis Expressing Key Pathways of Human Pathology

**DOI:** 10.3390/cells9092014

**Published:** 2020-09-01

**Authors:** Anita M. van den Hoek, Lars Verschuren, Nicole Worms, Anita van Nieuwkoop, Christa de Ruiter, Joline Attema, Aswin L. Menke, Martien P. M. Caspers, Sridhar Radhakrishnan, Kanita Salic, Robert Kleemann

**Affiliations:** 1Department of Metabolic Health Research, The Netherlands Organization for Applied Scientific Research (TNO), 2333 CK Leiden, The Netherlands; nicole.worms@tno.nl (N.W.); anita.vannnieuwkoop@tno.nl (A.v.N.); christa.deruiter@tno.nl (C.d.R.); joline.attema@tno.nl (J.A.); aswin.menke@tno.nl (A.L.M.); kanita.salic@tno.nl (K.S.); robert.kleemann@tno.nl (R.K.); 2Department of Microbiology and Systems Biology, The Netherlands Organization for Applied Scientific Research (TNO), 3704 HE Zeist, The Netherlands; lars.verschuren@tno.nl (L.V.); martien.caspers@tno.nl (M.P.M.C.); 3Research Diets, Inc., New Brunswick, NJ 08901, USA; sridhar.radhakrishnan@researchdiets.com; 4Department of Vascular Surgery, Leiden University Medical Center, 2333 ZA Leiden (LUMC), The Netherlands

**Keywords:** NAFLD, NASH, inflammation, fibrosis, metabolic syndrome, atherosclerosis, animal model

## Abstract

Non-alcoholic steatohepatitis (NASH) is a fast-growing liver disorder that is associated with an increased incidence of cardiovascular disease and type 2 diabetes. Animal models adequately mimicking this condition are scarce. We herein investigate whether Ldlr−/−. Leiden mice on different high-fat diets represent a suitable NASH model. Ldlr−/−. Leiden mice were fed a healthy chow diet or fed a high-fat diet (HFD) containing lard or a fast food diet (FFD) containing milk fat. Additionally, the response to treatment with obeticholic acid (OCA) was evaluated. Both high-fat diets induced obesity, hyperlipidemia, hyperinsulinemia, and increased alanine aminotransferase (ALT) and aspartate aminotransferase (AST) levels. Mice on both diets developed progressive macro- and microvesicular steatosis, hepatic inflammation, and fibrosis, along with atherosclerosis. HFD induced more severe hyperinsulinemia, while FFD induced more severe hepatic inflammation with advanced (F3) bridging fibrosis, as well as more severe atherosclerosis. OCA treatment significantly reduced hepatic inflammation and fibrosis, and it did not affect atherosclerosis. Hepatic transcriptome analysis was compared with human NASH and illustrated similarity. The present study defines a translational model of NASH with progressive liver fibrosis and simultaneous atherosclerosis development. By adaptation of the fat content of the diet, either insulin resistance (HFD) or hepatic inflammation and fibrosis (FFD) can be aggravated.

## 1. Introduction

Non-alcoholic steatohepatitis (NASH) is a progressive liver disease and a severe form of non-alcoholic fatty liver disease (NAFLD) that has become one of the most prevalent forms of chronic liver disease worldwide. NASH is characterized by the accumulation of fat in the liver (steatosis) in concert with inflammation, which can progress to liver fibrosis and cirrhosis. The disease is closely associated with obesity, insulin resistance, and dyslipidemia and due to the worldwide epidemic of obesity and the abundance of energy-dense foods with high levels of sugars and saturated fat, the prevalence of NAFLD/NASH continues to rise. The presence and severity of NAFLD is related to increased long-term morbidity and mortality. Particularly, liver fibrosis has been shown to be a strong predictor for NAFLD-related mortality [1,2]. Although NAFLD-related mortality can be due to adverse hepatic outcomes, such as cirrhosis, liver failure, and hepatocellular carcinoma, the primary cause of mortality in patients with NASH is cardiovascular disease (CVD) [3]. As an approved therapy for NAFLD/NASH is still lacking, there exists a great need to examine the effects of new therapeutic strategies on NASH more comprehensively, i.e., also taking into account cardiovascular endpoints such as atherosclerosis. Ideally, novel NASH therapeutics would have additional favorable effects on CVD risk, on top of their beneficial hepatic effects.

The development of novel NASH therapeutics entails an animal model that mimics the human disease state. Various animal models for NASH have been proposed [4,5]. Most of them are mainly based on the development of histopathological features but without characterizing underlying molecular pathways. Often, they have insufficient translatability as they do not have the metabolic syndrome-like profile associated with human NASH. In addition, most animal models lack sufficient validation by comparing effects of pharmacological interventions to those seen in humans. Hence, the translational value of most models remains debated. A recent study compared various animal models with NAFLD patients of different stages using gene profiling and concluded that there was very little overlap in the underlying disease pathways [6]. However, it was shown that high-fat diet-induced models were associated more closely with human fatty liver disease than other models. In general, it can be said that diet-induced rodent models better reflect the histopathological and molecular characteristics of human NASH [7,8,9]. However, within the category of diet-induced models, great differences exist regarding the translatability of the diet used, the recapitulation of molecular disease mechanisms, the development and severity of insulin resistance and fibrosis, as well as the development of cardiovascular endpoints, which entirely depends on genetic predisposition.

In the current study, we investigated whether Ldlr−/−.Leiden mice, which are genetically predisposed to develop CVD and that have been extensively characterized for recapitulating features of metabolic syndrome and NASH [7,10,11,12,13,14,15,16,17], would represent a translational model in which advanced stages of liver fibrosis can be studied in conjunction with atherosclerosis. Ldlr−/−.Leiden mice display histopathological characteristics of NASH, in the context of obesity, hyperlipidemia, and hyperinsulinemia, when fed energy-dense diets with macronutrient composition comparable to human diets [14]. Importantly, this model does not rely on diets with amino acid and choline deficiency or the supplementation of cholesterol (a requirement for many NASH models). This model, although not included in the original comparison of hepatic gene expression in murine and human NASH described above [6], showed more overlap in underlying disease pathways in a comparison with the same human gene profiling dataset [7]. In the current study, we tested whether adaptations regarding the fat and carbohydrate source of the NASH-inducing diet would aggravate the severity of liver fibrosis and NASH-associated comorbidities, i.e., insulin resistance and atherosclerosis. To this end, two obesogenic diets with macronutrient composition akin to human diets were used that differed in their fat (lard or milk fat) and carbohydrate (sucrose or fructose) sources. Features of the metabolic syndrome, NASH, hepatic fibrosis, and atherosclerosis were evaluated using multiple histological and biochemical techniques. Translatability was assessed by liver gene expression profiling and comparison to human NASH. Furthermore, we investigated within this model the response to the therapeutic administration of OCA, one of the drugs currently in late stage development for the treatment of NASH.

## 2. Materials and Methods

### 2.1. Animals and Experimental Design

All animal care and experimental procedures were approved by the Ethical Committee on Animal Care and Experimentation (Zeist, The Netherlands; approval reference numbers TNO-129, Date: 8 March 2016 and TNO-261, Date: 6 April 2017) and were in compliance with European Community specifications regarding the use of laboratory animals. Male Ldlr−/−.Leiden mice (TNO, Metabolic Health Research, Leiden, The Netherlands) were used. This substrain of the Ldlr-/- mouse has a 94% C57BL/6J background and 6% 129S1/SvImJ background. Mice were group housed in a temperature-controlled room on a 12 h light–dark cycle and had free access to food and heat-sterilized water. For the first experiment, 11–17-week-old mice were matched on age, body weight, blood glucose, plasma cholesterol, and triglycerides into four groups of mice: mice (*n* = 10) that were kept on the healthy grain-based chow diet (R/M-H, Ssniff Spezialdieten GmbH, Soest, Germany), mice (*n* = 10) that were given a high-fat diet (HFD) containing 45 kcal% fat from lard, 35 kcal% from carbohydrates (primarily sucrose), and 20 kcal% casein (D12451, Research Diets, new Brunswick, NJ, USA) and mice (*n* = 8) that were given a fast food diet (FFD) containing 41 kcal% fat from milk fat, 44 kcal% from carbohydrates (primarily fructose), and 14 kcal% casein (Research Diets, new Brunswick, NJ, USA) for 28 weeks. Both the HFD and FFD were not supplemented with crystalline cholesterol but contain only trace amounts of this lipid in its naturally occurring form, since it is a constituent of lard and milk fat. Based on the natural cholesterol content of lard or milk fat, the HFD and FFD contained approximately 0.01% and 0.05% (*w/w*) cholesterol, respectively, which has been confirmed by biochemical analyses of HFD by others and us [15]. In addition, one group of mice (*n* = 8) was added that was given the same FFD for 28 weeks but was treated with the farnesoid X receptor (FXR) agonist obeticholic acid (OCA) (Bio-Connect, Huissen, The Netherlands) provided as diet admix (10 mg/kg/d) from week 18 to 28. Animals were sacrificed unfasted by gradual-fill CO_2_ asphyxiation in week 28. In a separate experiment, 17–20-week-old mice (*n* = 13) were fed the FFD for 22 weeks. At t=12 weeks, a caudate liver lobe was ligated using isoflurane anesthetics (Abbott Laboratories Ltd., Sittingbourne, UK) and 4% (*w/w*) lidocaine spray (Xylocaine^®®^ Spray, AstraZeneca, Cambridge, UK) with pre-operative (0.06 mg/kg, sc) fentanyl (Bipharma, Almere, The Netherlands), (0.2 mg/kg, sc) midazolam (Actavis, Hafnarfjordur, Iceland), and post-operative (5 mg/kg sc) carprofen rimadyl (Pfizer Animal Health B.V., Capelle aan den IJssel, The Netherlands) and at t = 22 weeks, mice were sacrificed unfasted using gradual-fill CO_2_ asphyxiation. In both experiments, body weight and food intake per cage were measured regularly during the study. Blood samples were taken from the tail vein after 5 h fasting (with food withdrawn around 08.00 h) in EDTA-coated tubes (Sarstedt, Nümbrecht, Germany). Terminal blood was collected through cardiac heart puncture to prepare EDTA plasma and livers and perigonadal, visceral, and subcutaneous white adipose tissue (WAT), were collected, weighed, and fixed in formalin, and paraffin-embedded (lobus sinister medialis hepatis and lobus dexter medialis hepatis, left side perigonadal and subcutaneous WAT and half of visceral WAT) for histological analysis or (remaining liver lobes) fresh-frozen in N_2_ and subsequently stored at −80 °C for biochemical analysis and gene expression analysis. Hearts with aortic root area were collected, formalin-fixed, paraffin-embedded, and used for histological analysis of atherosclerosis development.

### 2.2. Plasma and Liver Biochemical Analysis

Blood glucose was measured at the time of blood sampling using a hand-held glucometer (Freestyle Disectronic, Vianen, The Netherlands). Plasma cholesterol and triglycerides were determined using enzymatic assays (CHOD-PAP and GPO-PAP, respectively; Roche Diagnostics, Almere, The Netherlands). Plasma insulin was analyzed by ELISA (Mercodia AB, Uppsala, Sweden). Homeostasis model assessment (HOMA) was used to calculate relative insulin resistance (IR). Five hours fasting plasma insulin and fasting blood glucose values were used to calculate IR, as follows: IR = [insulin (ng/mL) × glucose (mM)]/22.5. Plasma alanine aminotransferase (ALT) and aspartate aminotransferase (AST) were measured using a spectrophotometric activity assay (Reflotron-Plus, Roche). Serum amyloid A (SAA) was analyzed by ELISA (Tridelta, Maynooth, Ireland) and both E-selectin and monocyte chemoattractant protein-1 (MCP-1) were analyzed using Duoset mouse ELISAs (R&D Systems, Minneapolis, Canada) according to the manufacturer’s instruction. Hepatic collagen content was measured via a hydroxyproline-based colorimetric assay as a marker of fibrosis using the Sensitive total collagen assay (Quickzyme, Leiden, The Netherlands).

### 2.3. Histology

Liver samples (lobus sinister medialis hepatis and lobus dexter medialis hepatis) were collected (from non-fasted mice), fixed in formalin, and paraffin-embedded, and 3 µm sections were stained with hematoxylin and eosin (H&E) and Sirius Red. NASH was scored blindly by a board-certified pathologist in H&E stained cross-sections using an adapted grading system of human NASH [18,19]. In short, the level of macrovesicular and microvesicular steatosis was determined at 40× to 100× magnification relative to the total liver area analyzed and expressed as a percentage. Inflammation was scored by counting the number of aggregates of inflammatory cells per field using a 100× magnification (view size of 4.2 mm^2^). The averages of five random non-overlapping fields were taken, and values were expressed per mm^2^. Hepatic fibrosis was identified using Sirius Red stained slides and evaluated by computerized image analysis of hepatic collagen content (as percentage of liver surface area and including blood vessels). In addition, a qualitative analysis regarding the fibrosis stage was performed by a certified pathologist using the protocol of Tiniakos et al. [20], in which the presence of pathological collagen staining was scored as either absent (F0), observed within perisinusoidal/perivenular or periportal area (F1), within both perisinusoidal and periportal areas (F2), bridging fibrosis (F3) or cirrhosis (F4). Oxidative stress-related marker 4-hydroxynonenal (4-HNE) was analyzed in 4-HNE-stained liver sections (using overnight incubation at 4 °C with Rabbit anti-4-HNE Michael adducts primary antibody (1:1000 in PBS, ref.393207, Millipore Corporation, Billerica, MA, USA PBS), as previously described [21]) and by scoring the average 4-HNE positive immunoreactivity in five random non-overlapping fields and expressed per mm^2^.

Perigonadal WAT samples were collected, fixed in formalin, and paraffin-embedded, and 5 µm sections were stained with hematoxylin–phloxine–saffron (HPS), and adipose tissue inflammation was measured by counting crown-like structures (CLS) per field using a 100× magnification, and values were expressed per mm^2^.

Hearts were fixed in formalin, embedded in paraffin, and sectioned perpendicular to the axis of the aorta. Serial cross-sections (5 μm thick with intervals of 50 μm) were stained with HPS for histological analysis. Then, the average total lesion area per cross-section was calculated [22,23]. For determination of lesion severity, the lesions were classified into five categories according to the American Heart Association classification [24]: (0) no lesion, (I) early fatty streak, (II) regular fatty streak, (III) mild plaque, (IV) moderate plaque, and (V) severe plaque.

### 2.4. Transcriptome Analysis

Nucleic acid extraction was performed as described previously in detail [25]. Total RNA was extracted from individual lobus dexter lateralis samples using glass beads and RNA-Bee (Campro Scientific, Veenendaal, The Netherlands). RNA integrity was examined using the RNA 6000 Nano Lab-on-a-Chip kit and a bioanalyzer 2100 (Agilent Technologies, Amstelveen, The Netherlands). The NEBNext Ultra II Directional RNA Library Prep Kit (NEB #E7760S/L, New England Biolabs, Ipswich, MA, USA) was used to process the samples. Briefly, mRNA was isolated from total RNA using the oligo-dT magnetic beads. After fragmentation of the mRNA, cDNA synthesis was performed, and cDNA was ligated with the sequencing adapters and amplified by PCR. The quality and yield of the amplicon were measured (Fragment Analyzer, Agilent Technologies, Amstelveen, The Netherlands) and were as expected (broad peak between 300–500 bp) and 1.6 pM amplicon was used for sequencing. RNA expression was determined by RNA sequencing using the Illumina Nextseq 500 according to Illumina’s protocol by the service provider GenomeScan B.V (Leiden, the Netherlands) yielding at least 15 million reads per sample, 75nt single-end reads. The genome reference and annotation file Mus_musculus.GRCm38.gencode.vM19 was used for analysis in FastA and GTF format. The reads were aligned to the reference sequence using the STAR 2.5 algorithm with default settings (https://github.com/alexdobin/STAR). Based on the mapped read locations and the gene annotation, HTSeq-count version 0.6.1p1 was used to count how often a read was mapped on the transcript region. These counts serve as input for the statistical analysis using the DEseq2 package [26]. Selected differentially expressed genes (DEGs), corrected for multiple testing, were used as an input for pathway analysis (*p*-adjusted < 0.01) through the Ingenuity Pathway Analysis suite (www.ingenuity.com, accessed 2020).

To evaluate the representation of human pathophysiological pathways in HFD- and FFD-fed Ldlr−/−.Leiden mice, murine hepatic gene expression profiles were compared with published data on hepatic gene expression profiles in human NASH: a disease signature for NASH patients versus control [6,27] that was downloaded from the Gene Expression Omnibus (GEO) with accession number GSE48452 and a gene profile that differentiates NASH patients with severe fibrosis (fibrosis stage F3 or 4) from NASH patients with mild fibrosis (fibrosis stage F0 or 1) (GEO accession number GSE31803) [28].

### 2.5. Statistical Analysis

All values shown represent means ± SEM. Statistical differences between groups were determined by using non-parametric Kruskal–Wallis followed by a Mann–Whitney U test for independent samples using SPSS software. A *p*-value < 0.05 was considered statistically significant. Two-tailed *p*-values were used. In the case of transcriptome analysis, we selected differentially expressed genes using *p*-values, adjusted for multiple testing (False Discovery Rate, FDR) < 0.01 AND abs2logRatio > 0.5. The differentially expressed pathways (DEP) were selected based on Fischer’s exact test in the Ingenuity Pathway Analysis Software.

## 3. Results

### 3.1. HFD and FFD Induce Features of the Metabolic Syndrome in Ldlr−/−.Leiden Mice

Ldlr−/−.Leiden mice fed a HFD or FFD for 28 weeks developed on both diets pronounced obesity compared to age-matched control mice fed a low-fat chow diet, while food intake was similar (Table 1). Plasma insulin levels continued to rise during the study on the HFD and were significantly increased as compared to the chow diet at t = 28 (5.1-fold, *p* < 0.001), while glucose levels remained similar, resulting in a significantly higher insulin resistance on the HFD (4.8-fold increase in HOMA-IR at t = 28, *p* = 0.001). In case of the FFD, plasma insulin levels significantly increased until t = 18 (Appendix A) and then slowly decreased again, resulting in non-significant insulin levels at t = 28 as compared to the chow diet (Table 1). The FFD led to slightly but significantly lower glucose levels (−15% at t = 28, *p* = 0.006), resulting in a similar HOMA-IR index as on the chow diet. In response to both diets, mice developed hypercholesterolemia and severe hypertriglyceridemia (cholesterol: 4.0-fold and 5.2-fold increase versus chow diet, both *p* < 0.001, for HFD and FFD, respectively: triglycerides: 4.3-fold, *p* = 0.002 and 5.6-fold, *p* < 0.001 increase versus chow diet for HFD and FFD, respectively). Markers of liver injury, plasma ALT, and AST were significantly elevated on both diets as compared to the chow diet (ALT: 6.8-fold, *p* < 0.01 and 6.5-fold, *p* < 0.001, for HFD and FFD, respectively; AST: 3.9-fold and 5.1-fold, both p < 0.001, for HFD and FFD, respectively). Inflammation markers serum amyloid A (SAA), E-selectin, and monocyte chemoattractant protein-1 (MCP-1) were all significantly elevated on both diets as compared to the chow diet (SAA: 6.2-fold and 8.1-fold, both *p* < 0.001, for HFD and FFD, respectively; E-selectin: 1.4-fold, *p* < 0.05 and 1.4-fold, *p* < 0.001, for HFD and FFD, respectively; MCP-1: 2.3-fold and 2.1-fold, both *p* < 0.01, for HFD and FFD, respectively).

### 3.2. HFD and FFD Differentially Affect WAT

Although both HFD and FFD induced a similar obesity in Ldrl−/−.Leiden mice, the diets resulted in slightly different effects on the various adipose tissue depots (Figure 1A–C). After 28 weeks of diet feeding, both the HFD and FFD led to a significant increase in subcutaneous fat weight as compared to the chow fed mice (both 3.2-fold increase, *p* < 0.001). However, the HFD resulted in a significant increase in visceral (or mesenteric) fat weight versus mice on chow diet (1.4-fold increase, *p* = 0.004), while perigonadal (or epidydimal) fat weight was comparable to chow. In contrast, 28 weeks of FFD resulted in a significant increase in perigonadal fat weight versus mice on chow diet (1.6-fold increase, *p* = 0.003), while visceral fat weight was similar, suggesting differential effects on WAT expansion dynamics. While HFD did not result in a larger expansion of the perigonadal adipose tissue depot as compared to the expansion of aging mice on the chow diet, the perigonadal fat depot had significantly more inflammation on the HFD, as shown by the increased (3.8-fold, *p* = 0.001) number of crown-like structures (CLS)/mm^2^ perigonadal WAT (Figure 1C). For the FFD, and on top of the larger size of the perigonadal fat depot, also the number of CLS/mm^2^ tended to be increased as compared to the chow diet (2.2-fold, *p* = 0.052).

### 3.3. Ldlr−/−.Leiden Mice Develop Steatosis, Hepatic Inflammation, and Fibrosis.

Both HFD and FFD increased liver weight (both absolute values and relative to body weight; Table 1) and induced pronounced macro- and microvesicular steatosis, hepatic inflammation, and fibrosis after 28 weeks of diet feeding in Ldlr−/−.Leiden mice (Figure 2A). While the increase in microvesicular steatosis (Figure 2C) was similar on both diets (5.9-fold, *p* < 0.001 and 5.2-fold, *p* = 0.002 versus chow for HFD and FFD, respectively), the increase in macrovesicular steatosis (Figure 2B) was more pronounced for FFD as compared to HFD (8.6-fold and 11.6-fold increase versus chow, both *p* < 0.001, for HFD and FFD, respectively; 1.4-fold increase, *p* = 0.003 FFD versus HFD). Both diets induced hepatic inflammation, characterized by the presence of mononuclear cells and polymorphonuclear cells that formed aggregates, although the induction of hepatic inflammation (Figure 2D) was more pronounced for FFD as compared to HFD (19.0-fold and 35.9-fold increase versus chow diet for HFD and FFD, respectively, both *p* < 0.001; 1.9-fold increase, *p* = 0.027 FFD versus HFD).

However, the largest difference in induction was found for hepatic fibrosis, which was measured by the computerized analysis of collagen deposition in histological slices (Figure 2E) or biochemical collagen measurement (Figure 2F). Both diets led to a significant increase versus chow diet in histological collagen deposition (2.0-fold, *p* = 0.005 and 7.6-fold, *p* < 0.001, for HFD and FFD, respectively) or biochemical analyzed collagen content (3.9-fold and 8.7-fold, both *p* < 0.001, for HFD and FFD, respectively). Interestingly, the FFD clearly led to a markedly higher fibrosis induction for both parameters than the HFD (3.9-fold and 2.2-fold, both *p* < 0.001, for histological collagen deposition and biochemical collagen analysis, respectively). Importantly, the biochemical quantification demonstrated that, also in absolute amounts, FFD-induced collagen deposition was high (and exceeded 30 ug/mg liver protein). Fibrosis evaluation by a pathologist revealed that for the HFD, fibrosis was primarily located within perisinusoidal and/or periportal area (score F1–F2) with occasionally bridging fibrosis (F3), while for the FFD, all mice except one (F2) had bridging fibrosis (F3).

As the FFD led to a more severe NASH and fibrosis induction than the HFD, we next examined the disease induction on the FFD at earlier time points by feeding Ldlr−/−.Leiden mice the FFD for 22 weeks and taking a liver biopsy of the same mice at t = 12 weeks. By comparing the different NASH characteristics of the 12 and 22 week time points with those of the 28 weeks chow and FFD groups, it became evident that both macro- and microvesicular steatosis (Figure 3A,B) are already significantly increased as compared to the chow diet early in time (9.0-fold, *p* < 0.001 and 4.6-fold, *p* = 0.003 at t = 12 weeks, respectively). Hepatic inflammation (Figure 3C) revealed a different dynamical pattern with a gradual increase up until 22 weeks followed by a decrease in week 28. Nevertheless, despite this dynamic, hepatic inflammation in the FFD group was at all time points significantly higher as compared to the chow group. Hepatic fibrosis (Figure 3D) increased more gradually over time with a significant difference as compared to chow at 22 weeks (4.0-fold, *p* < 0.001). At 22 weeks, fibrosis was primarily located within the perisinusoidal and/or periportal area (F1–F2) with occasionally bridging fibrosis (F3), while after 28 weeks of FFD all mice except one (F2) had bridging fibrosis (F3).

To validate whether the Ldlr−/−.Leiden mice on FFD respond to treatment with pharmaceutical intervention, Ldlr−/−.Leiden mice were fed the FFD supplemented with an established [11] and relatively low dose of OCA (10 mg/kg/d) from week 18 to 28. OCA treatment did not significantly affect body weight (50.4 ± 2.6 g), food intake (14.0 ± 0.4 kCal/mouse/day), or liver weight (4.0 ± 0.5 g) as compared to the untreated FFD fed animals (Table 1). The intervention did not affect macro- and microvesicular steatosis (Figure 3A,B), but it significantly decreased hepatic inflammation (with −59%, *p* = 0.038; Figure 3C) and hepatic fibrosis (with −57%, *p* = 0.028; Figure 3D).

### 3.4. Ldlr−/−.Leiden Mice on HFD and FFD Recapitulate Human NASH and Fibrosis Pathways

To further investigate the mechanisms and pathways modulated by the different diets, pathways that were differentially expressed in the liver by HFD or FFD relative to chow were compared in the first study. While HFD led to a total of 187 differentially expressed pathways, FFD led to a larger amount of differentially expressed pathways (*n* = 237). The majority of the pathways affected overlapped (*n* = 167), and only a relatively small portion was unique for HFD (*n* = 20) or for FFD (*n* = 70; see Venn diagram, Figure 4A).

The shared top 10 most enriched pathways for HFD and FFD are shown in Figure 4B and contain important pathogenic pathways such as ‘Fibrosis/Stellate cell activation’, metabolic pathways such as ‘LXR/RXR Activation’, and ‘GP6 Signaling Pathway’ and predominantly inflammatory pathways. Almost all (166 out of 167) of the overlapping, shared pathways were regulated in the same direction by both diets. For the 20 pathways uniquely affected by HFD, the direction of regulation was the same as with FFD, but the FFD effect was less pronounced and did not reach statistical significance. The top 3 pathways of this portion of the Venn diagram were pathways critical for cholesterol biosynthesis (‘Cholesterol biosynthesis III, II and I’ were more downregulated on the HFD as compared to FFD). For the 70 pathways uniquely affected by FFD, the direction of most pathways (68 out of 70) was the same with HFD, but HFD effects were more modest and not significant. A most remarkable difference between FFD and HFD was revealed by the two most enriched pathways of this specific portion of the Venn diagram: ‘Oxidative Phosphorylation’ and ‘Mitochondrial Dysfunction’, both of which were uniquely affected by FFD. The pathway ‘Oxidative phosphorylation’ is a section of the broader pathway ‘Mitochondrial Dysfunction’, and although clearly downregulated by both diets, more genes were downregulated on the FFD, specifically in the different complexes of the electron transport chain and ultimately resulting in the downregulation of ATP synthases with FFD only (Figure 4C), indicating that FFD has a greater detrimental effect on mitochondrial function than HFD. This notion was further substantiated by measuring hepatic 4-hydroxynonenal (4-HNE), a marker for oxidative stress, and we found a significant upregulation of this marker for HFD and FFD as compared to the chow diet (33-fold, *p* < 0.001 and 57-fold, *p* = 0.003, respectively; Figure 4D).

Since a remarkable histological difference between the diets was the stronger induction of hepatic inflammation and hepatic fibrosis on the FFD as compared to the HFD, we analyzed the pathways and genes involved in inflammation and fibrosis in more detail. Most of the inflammatory pathways and underlying genes were upregulated by both diets but showed a more pronounced upregulation with the FFD (Figure 4E). In addition, we analyzed the transcriptome dataset for their cell-type specific gene expression profiles of leukocytes essentially as reported by Palmer et al. [29]. These cell-type specific gene sets allow an accurate assessment of the presence of particular leukocytes in a tissue. We found higher −log (*p*-values) for the B-cell population and T-cell population on the high fat diets versus chow (B-cell population: HFD versus chow: −log (*p*-value) = 9.8; FFD versus chow: −log (*p*-value) = 6.7; T-cell population: HFD versus chow: −log (*p*-value) = 3.5; FFD versus chow: −log (*p*-value) = 5.0). This supports the notion that the inflammatory aggregates consist of different inflammatory cells. By comparing the hepatic gene expression profiles of the mice on the two different high-fat diets with a reported human-based disease signature that differentiates NASH patients from normal controls (Figure 4F), we could investigate the representation of these pathophysiological pathways on both diets. Ldlr−/−.Leiden mice on both HFD and FFD displayed a gene expression profile that for the majority of the genes was similar to the human-based NASH gene set [6,26]. Based on Fischer’s exact test, both HFD (−log(*p*val) = 14.4) and FFD (−log(*p*val) = 11.0) showed a significant enrichment for the Teufel dataset. For both diets, only a small subset of genes revealed a different regulation pattern, and this subset was similar for both diets. Furthermore, a second independent human gene set that is upregulated in NASH patients with severe fibrosis (stage F3 or 4) and distinguishes them from NASH patients with mild fibrosis (stage F0 or 1) was used. This gene set was enriched significantly as well for the majority of those genes in the Ldlr−/−.Leiden mice on both HFD (−log(*p*val) = 13.9) and FFD (−log(*p*val) = 12.8) (Appendix A) [27]. For both human gene sets, the gene regulation was stronger for most genes in the mice on the FFD as compared to the HFD.

Perhaps even more important than similarity on individual gene level is whether the mice on HFD or FFD mimic human NASH on a pathway level. Thereto, the differentially expressed pathways that distinguish NASH patients from controls were analyzed for the mice on both diets as well. Of the in total 65 differentially expressed pathways identified in humans, 80% were recapitulated on FFD and 73% were recapitulated on HFD. In total, 15% were not represented in mouse livers collected after 28 weeks of diet-feeding and 68% were represented by both diets (Figure 4G). Among the pathways that were represented by both diets were important pathways such as hepatic fibrosis/stellate cell activation, several inflammatory pathways, and also atherosclerosis signaling. For the complete list of pathways and representation thereof on the HFD and FFD, see Appendix A.

In aggregate, these data indicate that Ldr-/-.Leiden mice on either HFD or FFD recapitulated most of the pathophysiological pathways identified in human NASH and fibrosis. Gene expression changes on HFD were very similar to those observed with FFD, but the effects of FFD were in general stronger. In addition, FFD affected more pathways than HFD, including a pronounced downregulation of the ATP synthases in the ‘Oxidative Phosphorylation’ pathway.

### 3.5. Ldlr−/−.Leiden Mice Develop Atherosclerosis

Ldlr−/−.Leiden mice are hyperlipidemic mice that are known to develop atherosclerosis on the HFD [13], which can be attenuated by different interventions [13,15]. To evaluate whether Ldlr−/−.Leiden mice develop atherosclerosis as well on the FFD, we next analyzed atherosclerosis development in the aortic valve area after 22 and 28 weeks. FFD induced pronounced atherosclerosis development with a total lesion area of 633.037 µm^2^ after 22 weeks and 889.430 µm^2^ after 28 weeks in the FFD group versus 86.589 µm^2^ in the chow group and 409.530 µm^2^ in the HFD group after 28 weeks (Figure 5A,B). Next to the lesion area, lesion severity was evaluated as well. Lesion severity significantly shifted on the FFD toward less mild lesions (2.7% after 22 weeks on FFD and 4.6% after 28 weeks on FFD versus 60.1% after 28 weeks on chow and 27.9% after 28 weeks on HFD) and more severe lesions (97.3% and 95.4% after 22 and 28 weeks, respectively, on FFD versus 39.9% after 28 weeks on chow and 72.1% after 28 weeks on HFD; Figure 5C). Treatment from 18 to 28 weeks with a relative low dose of OCA did not affect the atherosclerotic lesion area (930.743 µm^2^) in the Ldlr−/−.Leiden mice on FFD, nor lesion severity (1.0% mild and 99.0% severe lesions; Figure 5A–C).

## 4. Discussion

Nowadays, finding a reliable animal model of NASH that properly mimics the human disease state and the underlying mechanisms is still a scientific challenge. In the present study, we investigated whether Ldlr−/−.Leiden mice on different high-fat diets represent a suitable model exhibiting features of metabolic syndrome and cardiovascular disease, as they are observed in many NASH patients. More specifically, we investigated the biochemical and histopathological characteristics of these mice on two different high-fat diets (HFD, FFD) and demonstrated the major clinical characteristics of human pathology: metabolic syndrome with pronounced and stable hypertriglyceridemia and hyperinsulinemia, hepatic steatosis, hepatic inflammation, and fibrosis as well as atherosclerosis are all exhibited in the mice, with more pronounced pathology seen in FFD. Furthermore, we investigated the response of the FFD model to OCA, which is one of the drugs currently in late-stage development for the treatment of NASH.

Our first goal was to develop an animal model for NASH that displays a metabolic context resembling the human situation as closely as possible—that is, the presence of obesity and metabolic syndrome. For this reason, a transgenic mouse model was chosen: Ldlr−/−.Leiden mice, a specific substrain of conventional Ldlr-/- mice that are hyperlipidemic with a lipoprotein profile resembling humans and prone to develop obesity-related disorders when fed a high-fat diet. Two different high-fat diets were used to represent the obesogenic diets to which many subjects (including NASH patients) in modern societies are exposed to, and to induce obesity and insulin resistance in this model. Importantly, both diets had a macronutrient composition akin to human diets [30], were not deficient in essential nutrients or dietary components (such as e.g., methionine–choline-deficient diets), and were not supplemented with supraphysiological amounts of cholesterol (in contrast to most high fat-based models). Altogether, this resulted in dietary macronutrient and cholesterol levels that are translational to the human situation. The diets used primarily differed in their fat source (lard fat versus milk fat) and carbohydrate content (sucrose versus fructose). Both high fat diets induced hyperlipidemia, with elevated plasma cholesterol levels as well as robustly elevated triglyceride levels (the latter often missing in other NASH mouse models). Since Ldlr−/−.Leiden mice have a human-like lipoprotein profile, the elevated cholesterol levels are translational and due to an elevation of very-low density lipoprotein (VLDL) and primarily low-density lipoprotein (LDL) cholesterol instead of elevated high density lipoprotein (HDL) cholesterol levels (in contrast to wild-type mice and many other mouse models). In addition, plasma ALT and AST levels, important biomarkers indicative of liver damage in NASH patients, were elevated by both diets. Furthermore, both diets induced hyperinsulinemia, although the dynamics of the response varied between the diets. The HFD rapidly increased insulin levels 5-fold (versus chow group) after 6 weeks already and kept increasing until 22 weeks, after which it stabilized until 28 weeks, while the FFD more gradually increased in time until 18 weeks on the diet and then started to decrease again until 28 weeks (Appendix A). The differential dynamics in fasting plasma insulin may be related to the differential effects on WAT. Using the same HFD as herein, we have shown that perigonadal WAT expands and becomes inflamed and insulin resistant relatively early in time (after about 12 weeks). This was paralleled by hyperinsulinemia and preceded hepatic inflammation and hepatic insulin resistance, which developed later in time (after 24 weeks) as shown by Mulder et al. [31,32]. In case of FFD, perigonadal WAT expansion appears to be slower, which is reflected by the lower absolute WAT mass and less inflammatory CLS as compared to HFD. On the contrary, hepatic inflammation was significantly higher with FFD compared to HFD, and the observed systemic hyperinsulinemia during FFD may be due to hepatic insulin resistance rather than WAT insulin resistance. In general, the high fat diet-fed Ldlr−/−.Leiden model can be considered pre-diabetic with insulin resistance depending on the diet more or less prominently present, but without further progression toward full-blown type II diabetes (i.e., loss of pancreatic insulin production) within the time frame investigated.

As a second, but equally important requirement, the histological phenotype of the Ldlr−/−.Leiden mice on the high-fat diets needed to resemble the progression of human NASH pathology. Both diets resulted in the development of the two key features of NASH, steatosis, i.e., macrovesicular steatosis and microvesicular steatosis, and lobular inflammation, as well as hepatic fibrosis. This is in clear contrast to wild-type mice that develop a much milder form of steatosis with less prominent hepatic inflammation and an absence of fibrosis on the same HFD [19,33]. There is practically no fibrosis development in wild-type mice fed high-fat diets without cholesterol supplementation. Lobular inflammation in the Ldlr−/−.Leiden model consisted of a mixed inflammatory cell type of mononuclear cells and polymorphonuclear cells that formed aggregates and that have previously been characterized in more detail and stained positive for F4/80 (macrophages), CD-4 (T helper cells), and CD-8 (cytotoxic T cells) [11], as well as MPO (neutrophils) [10]. In the Ldlr−/−.Leiden mice fed a HFD or FFD, hepatic fibrosis was present in a pattern characteristic for dietary induction and resembling the human situation (in contrast to the induction by chemical toxins). Fibrosis was situated in both the pericentral and perisinusoidal zones with progression to bridging fibrosis when the diet duration was prolonged. However, on the HFD, the severity in fibrosis induction was not equally consistent among mice, as can be expected based on previous studies in which we showed that pronounced fibrosis development on HFD requires more than 30 weeks [11]. This corroborates with the disease variability that has been reported in other diet-induced obese mouse models of NASH [34,35,36]. In contrast, on the FFD, the fibrosis induction was observed in all mice compared to chow, and fibrosis measured by several methods was also much more pronounced compared to HFD. Importantly, FFD induced very severe stages of fibrosis very rapidly (within 28 weeks), which has, to our knowledge, not been observed in other diet-inducible experimental models of liver fibrosis. While fibrosis was clearly present (especially on the FFD), hepatocyte ballooning, another hallmark of human NASH that is defined as enlarged and rounded hepatocytes with clear cytoplasm, was only occasionally observed on both diets. This is consistent with our and others’ previous findings that ballooned cells are only marginally observed in rodent models of NASH and are less prominently existing as compared to human NASH [19,37,38,39].

Transcriptomics analysis of liver tissues illustrated that Ldlr−/−.Leiden mice on both diets recapitulate the underlying molecular disease pathways and shared a relevant number of central pathways involved in NASH and fibrosis pathophysiology. Among the NASH pathways that were represented in the Ldlr−/−.Leiden mice were metabolism-related pathways, several inflammatory pathways, hepatic fibrosis/hepatic stellate cell activation, as well as type II diabetes mellitus signaling and atherosclerosis signaling. Although the gene expression in the liver for type II diabetes mellitus signaling was activated, in the current experiment mice remained insulin-resistant without progression toward type II diabetes. In contrast, the activated gene expression in the liver for atherosclerosis signaling coincided with actual atherosclerosis development in the aortic root sections of the mice and revealed severe plaque formation. We and others have shown that disease-related gene expression signaling pathways are typically activated prior to manifestation of the respective pathological manifestation, e.g., a fibrosis gene expression can already be observed after 12 weeks of HFD in the model used herein [40], and it is possible that after prolonged HFD feeding, the observed severe insulin resistance will further progress toward type II diabetes. For application in NAFLD/NASH research, the established experimental conditions may be particularly relevant because of the observed high similarity to human NASH gene expression profiles obtained from two independent human studies, each of which emphasizes a different stage of NASH (in Teufel et al. [6], patients with more mild fibrosis stages were overrepresented and Moylan et al. [28] identified a gene set using severe F3/4 disease stage that was indicative for rapid progressors from mild to severe fibrosis). The liver gene expression of HFD and FFD-treated mice was obtained after one single time period of diet feeding (28 weeks), thus representing a snapshot in time with a relatively homogenous disease stage. It is remarkable that our transcriptomics profile strongly overlapped with the two human NASH gene sets, the more so because other diet-inducible models such as HFD-induced BL/6 mice, Western-type diet (cholesterol)-treated mice, the methionine-choline deficient (MCD) diet model, etc., did not recapitulate the same human gene sets, even when tested at two different time points [6].

In order to assess the value of the Ldlr−/−.Leiden model for compound testing, it is important to compare the effects of pharmacological treatments in mice to those observed in humans. In this study, we investigated the response to OCA under FFD conditions, because the effects of OCA under HFD conditions were reported in detail recently [11] and revealed a tendency to decrease macrovesicular steatosis and hepatic inflammation, while hepatic fibrosis development was attenuated (however, fibrosis was not reduced beyond the level at the start of intervention). In line with the observations in the Farnesoid X Receptor Ligand Obeticholic Acid in NASH Treatment (FLINT) trial [41], we observed in the present study an improvement of FFD-induced lobular inflammation and hepatic fibrosis, while we did not observe an improvement in steatosis. The observed effect of FFD on liver inflammation appears to be more pronounced than that of HFD, which is consistent with observations made earlier in a short-term diet intervention study in young mice using the same diets [14]. Our observation that more inflammatory aggregates were found at 18 weeks of FFD feeding compared to 28 weeks supports the view that hepatic inflammation is dynamic and that advanced stages of NASH may have less inflammation than earlier stages as reported by Koyama and Brenner [42]. Opposite to the effects in humans, we observed a reduction in plasma total cholesterol with OCA (data not shown), which is a well-known response in most preclinical rodent models and related to the higher bile acid synthesis in rodents [43]. Despite the lower cholesterol levels upon OCA treatment in the Ldlr−/−.Leiden mice, OCA did not affect atherosclerosis development in the aortic root section of the mice in the present study. The HFD-treated Ldlr−/−.Leiden mouse is sensitive to targets playing a role in human NASH including caspase-1 inhibitors, diglyceride acyltransferase (DGAT) inhibitors, sodium dependent bile acid transport inhibitors [10,16,44], as well as nutritional components such as butyrate, propionate, probiotica, and protein hydrolysates [13,45], but further validation studies of the model by evaluating the response with other NASH candidate drugs or established compounds in the future are warranted.

While on both high-fat diets the Ldlr−/−.Leiden mice develop NASH and atherosclerosis, adaptation of the diet led to either more pronounced insulin resistance (HFD) or more pronounced hepatic inflammation and fibrosis (FFD). The higher insulin resistance on the lard-containing HFD might be related to the difference in fat distribution. The perigonadal WAT is the first depot in mice that expands during HFD, and time course studies showed that after this depot has reached its maximal mass and expansion after 12 weeks (about 2.2–2.5 g in most mouse strains) [31], it starts to decline in weight (to about 1.8 g) while CLS continue to form resulting in a strongly inflamed tissue. In parallel with the decline of perigonadal WAT mass, excess energy is mainly stored in visceral or mesenteric WAT which then starts to expand, which is the condition at which the HFD-treated mice in the current study were sacrificed (at 28 weeks). The visceral WAT depot was significantly increased on the HFD relative to chow. As outlined before, the FFD is less obesogenic, potentially due to the slightly higher cholesterol content, and the perigonadal WAT depot was still in the expansion phase; thus, it significantly increased relative to chow. Besides this, the subcutaneous WAT, of which the expansion is also delayed compared to perigonadal WAT [31], was increased relative to chow with both diets. In particular, visceral mesenteric obesity has been associated with insulin resistance and therefore could explain the more pronounced insulin resistance with the HFD ([46,47] and references therein). A second contributing factor is the high inflammatory tone of perigonadal WAT in HFD-fed mice compared to FFD-fed mice, in which the inflammation is more liver centered. The more pronounced hepatic inflammation and associated fibrosis with the FFD might be related to the remarkable difference between both high-fat diets that became visible upon hepatic transcriptome analysis and revealed a downregulation with the FFD in mitochondrial complex V, which is involved in oxidative phosphorylation. Mitochondrial alterations are considered to be critical factors causing NAFLD and are thought to play an important role in promoting steatosis, inflammation, and progression to fibrosis [48,49]. Mitochondrial dysfunction can lead to an increased generation of reactive oxygen species (ROS) that could potentiate the effects of oxidative stress through the oxidization of polyunsaturated fatty acids, leading to the productions of aldehyde byproducts such as 4-hydroxynonenal (4-HNE; increased levels were confirmed in our study) and malondialdehyde (MDA) and also trigger pro-inflammatory cytokines ([50] and references therein).

Each animal model has its limitations. As it is presumed that human NASH has various disease etiologies, it is unlikely that a single mouse model can be used to investigate all pathogenic processes seen in patients. Obviously, the Ldlr−/−.Leiden mice cannot be used to investigate lipid-modulating interventions that rely on intact Ldl receptor function, such as statins, unless pleiotropic effects are investigated. However, for other metabolism-modulating interventions, incretins, peroxisome proliferator-activated receptor (PPAR) modulators, and anti-inflammatory or anti-fibrotic interventions, we expect that the model can be useful. As the Ldlr−/−.Leiden model develops the metabolic context with pronounced and stable hypertriglyceridemia and hyperinsulinemia, displays similar histological characteristics as well as the transcriptomic and metabolomics disease profiles [11] as NASH patients, develops atherosclerosis as well as gut dysfunction and dysbiosis [14], the model can be a valuable asset to study the disease comprehensively and complementary to other NASH models. This study emphasizes the importance of choosing the appropriate combination of model and diet for each research question, as it shows that by adaptations of the dietary constituents, different characteristics critical for human pathogenesis can be emphasized. Since the Ldlr−/−.Leiden mice develop NASH in association with atherosclerosis, the model is well suited to investigate the effects of new NASH therapeutics, alone (or drugs in combination) and evaluate simultaneously their effects on CVD.

## Figures and Tables

**Figure 1 cells-09-02014-f001:**
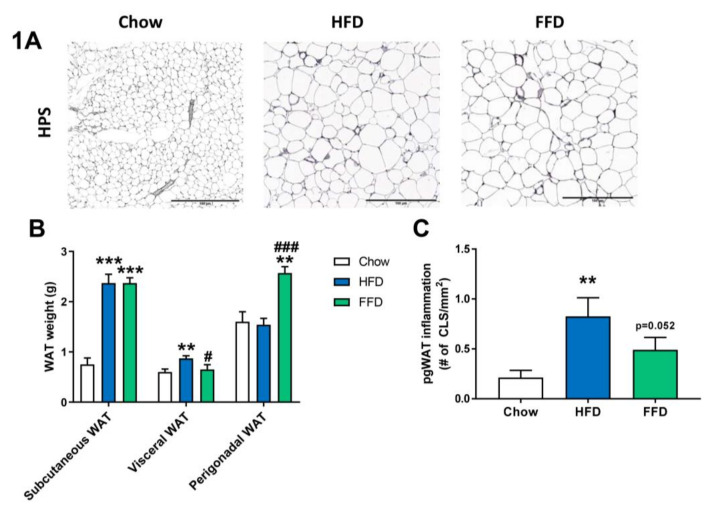
Representative images of perigonadal white adipose tissue (WAT) cross-sections (**A**) stained with hematoxylin–phloxine–saffron (HPS) and WAT weight of different adipose tissue depots (**B**) from Ldlr−/−.Leiden mice fed a healthy chow diet or fed a high-fat diet (HFD) containing lard or a fast food diet (FFD) containing milk fat for 28 weeks. Perigonadal WAT inflammation was analyzed by measuring the number of crown-like-structures (CLS)/mm^2^ (**C**). Values represent mean ± SEM for ≥8 mice per group. ** *p* < 0.01 and *** *p* < 0.001 vs. chow; # *p* < 0.05 and ### *p* < 0.001 FFD vs. HFD.

**Figure 2 cells-09-02014-f002:**
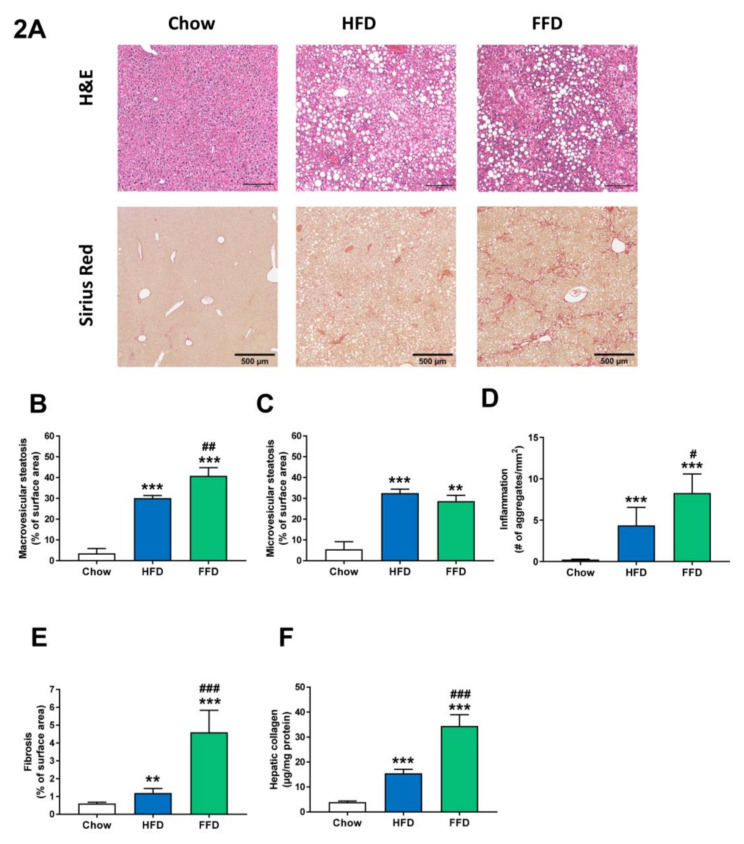
Representative images of liver cross-sections stained with hematoxylin and eosin (H&E) or Sirius Red (**A**) and quantitative analysis (**B**–**F**) from Ldlr−/−.Leiden mice fed a healthy chow diet or fed a high-fat diet (HFD) containing lard or a fast food diet (FFD) containing milk fat for 28 weeks. Macrovesicular (**B**) and microvesicular (**C**) steatosis as a percentage of total liver area, inflammation as number of inflammatory aggregates per mm^2^ microscopic field (**D**), and hepatic fibrosis were analyzed as percentage Sirius Red of surface area (**E**) or biochemically analyzed hepatic collagen/mg protein (**F**). Values represent mean ± SEM for ≥8 mice per group. ** *p* < 0.01 and *** *p* < 0.001 vs. chow; # *p* < 0.05, ## *p* < 0.01 and ### *p* < 0.001 FFD vs. HFD.

**Figure 3 cells-09-02014-f003:**
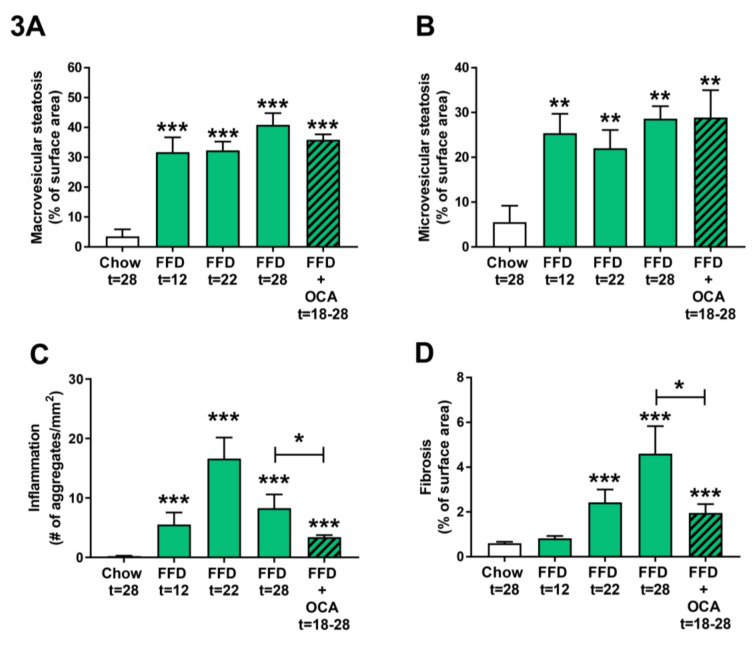
Quantitative analysis of non-alcoholic steatohepatitis (NASH) from Ldlr−/−.Leiden mice fed a healthy chow diet for 28 weeks or fed a fast food diet (FFD) containing milk fat for 12, 22, or 28 weeks. Data of FFD for 12 weeks were obtained by taking a liver biopsy of the mice that continued on the FFD for 22 weeks and are from a separate group of animals than mice that were fed the different diets for 28 weeks. In addition, mice were fed the FFD and treated with a relatively low dose of obeticholic acid (OCA) (10 mg/kg/d) from week 18 to 28. Macrovesicular (**A**) and microvesicular (**B**) steatosis as the percentage of total liver area, inflammation as the number of inflammatory aggregates per mm^2^ microscopic field (**C**), and hepatic fibrosis as the percentage of Sirius Red of surface area (**D**) were analyzed. Values represent mean ± SEM for ≥8 mice per group. ** *p* < 0.01 and *** *p* < 0.001 vs. chow or * *p* < 0.05 FFD + OCA vs. FFD t = 28 (as indicated by bar).

**Figure 4 cells-09-02014-f004:**
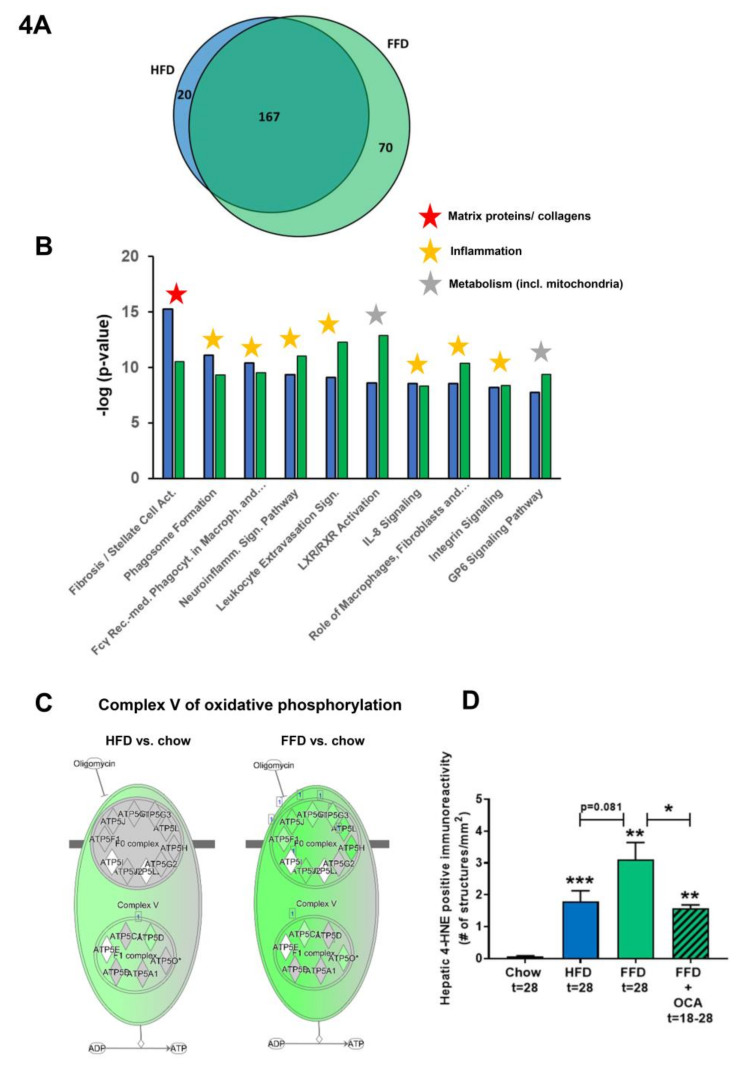
Venn diagram (**A**) illustrating the overlap of differentially expressed pathways (pathway *p*val < 0.01) of livers between Ldlr−/−.Leiden mice fed a high-fat diet (HFD) vs. healthy chow diet (blue circle) or fed a fast food diet (FFD) vs. healthy chow diet (green circle) for 28 weeks. N ≥ 8 mice per group. The shared top 10 of significantly enriched biological processes (−log (*p*-value)) for the genes affected by HFD (blue bars) and FFD (green bars) vs. chow (**B**). Differences between HFD and FFD in downregulation of ATP synthases in complex V of oxidative phosphorylation (**C**). Quantification of 4-hydroxynonenal (4-HNE) marker of oxidative stress using stained liver sections ** *p* < 0.01 and *** *p* < 0.001 vs. chow or * *p* < 0.05 FFD + OCA vs. FFD t = 28 (as indicated by bar) (**D**). Effect of HFD and FFD on different inflammatory pathways, expressed as median (2logR) of all genes in the listed inflammatory pathway (**E**). Heatmap showing expression of genes differentially regulated in human non-alcoholic steatohepatitis (NASH) vs. normal controls [6] in Ldlr−/−.Leiden mice fed the HFD or FFD for 28 weeks relative to chow (**F**). Blue color indicates downregulation, and red color indicates upregulation. Genes with # indicate genes with differential regulation vs. human regulation. Visualization (**G**) of the differentially expressed pathways (DEPs) distinguishing human NASH patients vs. normal controls and representation thereof in the mice on HFD or FFD.

**Figure 5 cells-09-02014-f005:**
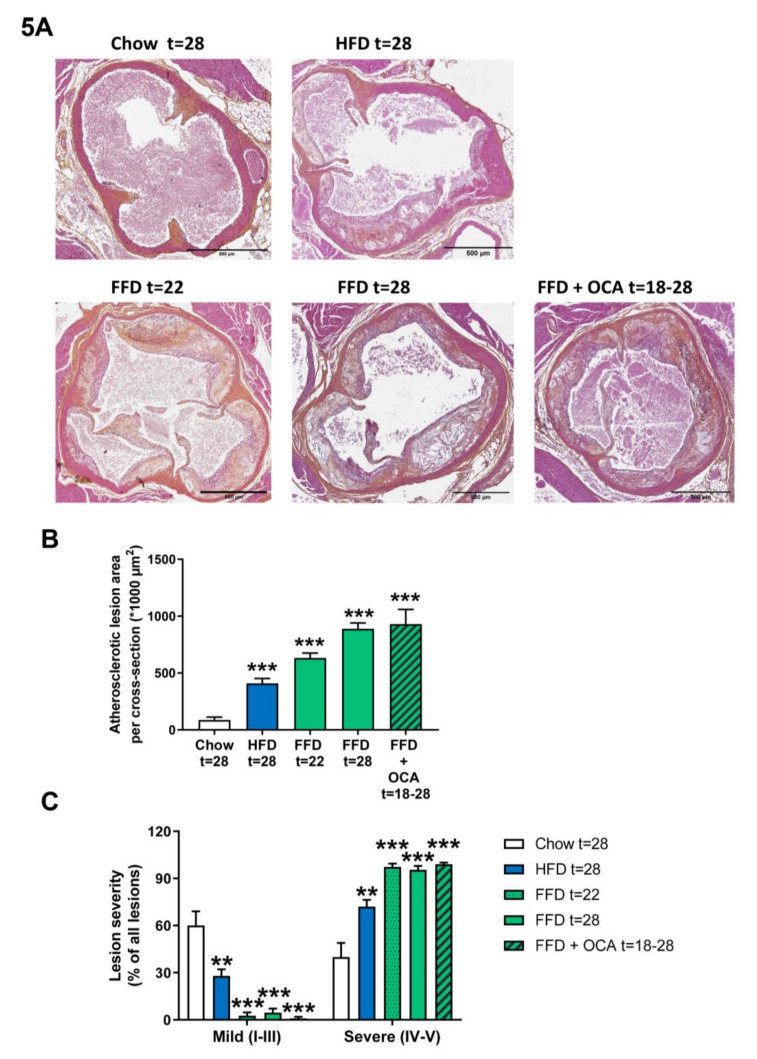
Representative images of atherosclerotic plaques in aortic root section (**A**) stained with hematoxylin–phloxine–saffron (HPS) from Ldlr−/−.Leiden mice fed a healthy chow diet for 28 weeks or fed a high-fat diet (HFD) for 28 weeks or fed a fast food diet (FFD) for 22 or 28 weeks. In addition, mice were fed the FFD and supplemented with a relatively low dose of OCA (10 mg/kg/d) from weeks 18 to 28. Total lesion area per cross-section was quantified (**B**) and lesion severity (**C**) was assessed, categorized as mild lesions (type I–III) and severe lesions (IV–V). Values represent mean ± SEM for ≥8 mice per group. ** *p* < 0.01 and *** *p* < 0.001 vs. chow.

**Table 1 cells-09-02014-t001:** Metabolic parameters.

	Chow	HFD	FFD
Body weight (g)	38.3 ± 1.5	52.3 ± 1.1 ***	48.7 ± 1.5 ***
Food intake (kCal/mouse/day)	13.5 ± 0.8	12.5 ± 0.6	14.9 ± 1.9
Liver weight (g)	1.8 ± 0.1	3.1 ± 0.2 ***	5.9 ± 0.4 *
Liver weight (% of body weight)	4.8 ± 0.2	3.5 ± 0.2 ***	7.1 ± 0.3 ***
Blood glucose (mM)	7.7 ± 0.4	7.7 ± 0.2	6.5 ± 0.3 **
Plasma insulin (ng/mL)	2.9 ± 0.6	14.7 ± 4.2 ***	3.9 ± 0.4
HOMA-IR	1.0 ± 0.2	4.9 ± 1.4 **	1.1 ± 0.1
Plasma cholesterol (mM)	8.0 ± 0.7	32.2 ± 3.7 ***	41.3 ± 4.7 ***
Plasma triglycerides (mM)	1.5 ± 0.3	6.5 ± 1.3 **	8.3 ± 1.2 ***
Plasma ALT (U/L)	53.3 ± 7.1	363.7 ± 64.8 **	348.6 ± 33.0 ***
Plasma AST (U/L)	88.2 ± 8.2	449.1 ± 69.1 ***	342.9 ± 58.4 ***
Plasma SAA (µg/mL)	39.6 ± 4.2	244.6 ± 54.6 ***	318.6 ± 73.3 ***
Plasma E-selectin (ng/mL)	61.3 ± 3.7	83.0 ± 10.1 *	83.7 ± 3.4 ***
Plasma MCP-1 (pg/mL)	33.2 ± 6.5	75.4 ± 11.2 **	70.7 ± 6.3 **

Ldlr−/−.Leiden mice were fed a healthy chow diet or fed a high-fat (HFD) diet containing lard fat or a fast food diet (FFD) containing milk fat for 28 weeks. Data represent mean ± SEM for *n* ≥ 8 mice/group. * *p* < 0.05, ** *p* < 0.01, *** *p* < 0.001 vs. chow. ALT: alanine aminotransferase, AST: aspartate aminotransferase, HOMA-IR: homeostasis model assessment of insulin resistance, MCP-1: monocyte chemoattractant protein-1, SAA: serum amyloid A.

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
