# Peer review of "A Translational Mouse Model for NASH with Advanced Fibrosis and Atherosclerosis Expressing Key Pathways of Human Pathology"

_cells, 2020, doi:10.3390/cells9092014_

Round 1
Reviewer 1 Report
The manuscript by Anita M. van den Hoek describes Ldlr-/-.Leiden mice on different high fat diets as new NASH model. The authors characterize the mice on chow fed, HFD, and FFD diet. Additionally, the authors investigated the response of obeticholic acid (OCA) on FFD mice. They also compared the transcriptome of the mice with the human NASH samples, showing overlap in the pathways between mice and human, suggesting the use of the Ldlr-/-.Leiden mice as a translational model for NASH.
One major issue I have with this study is there is no comparison with the WT mice. In parallel comparison with WT mice would have really strengthened the author’s claim of using the Ldlr-/- mice as a translational model for NASH. This is something the authors should address in discussion, as this is beyond the scope of the current manuscript.
Authors mention in the discussion that OCA study has been performed earlier, again a one line discussion of the findings and how it relates to the observation here would be useful.
In the revised version of the manuscript, I would like to see the following data:
- Liver wt/Body wt ratio comparison for the animals on chow vs HFD vs FFD diet.
- Authors quantify the inflammatory aggregates; however a characterization of the type of inflammatory cells is needed. Cd68, F4/80, MAC-2 staining as well as q-PCR analysis for other markers like Ly6C, MCP-1 is needed.
- Body weight as well liver wt/body wt ratio for mice with OCA treatment is required to show that the decrease in inflammation and fibrosis is not just due to eating less and loss in body weight.
- TUNNEL staining to look at apoptosis of hepatocytes with the different diets.
Author Response
We kindly thank you for the time and effort it took to review the paper.
Response to comparison with WT mice: An important reason for choosing Ldlr-/-.Leiden mice in our study was that we would like to mimic the metabolic syndrome like context seen in most NASH patients as closely as possible. An important factor with the metabolic syndrome is the presence of hyperlipidemia (besides obesity, insulin resistance etc). Wild-type mice have a completely different lipid metabolism. In contrast to humans, WT mice have a fast clearance of VLDL and LDL-particles and therefore the plasma cholesterol of WT mice primarily consists of HDL-cholesterol and the mice do not develop stable hypertriglyceridemia (nor develop atherosclerosis). Hence, hyperlipidemia in these mice is not translational to the human situation. In the Ldlr-/-.Leiden mice the diet-induced hyperlipidemia is characterized by an elevation of cholesterol confined to primarily LDL (and VLDL)-cholesterol and not to HDL-cholesterol. We have previously compared the Ldlr-/-.Leiden mice on the HFD with WT mice on HFD (see Mulder et al., Inflammation & Cell Signaling 2015; 2: e804. doi: 10.14800/ics.804 and Liang et al., PLoS One 2014 Dec 23;9(12):e115922. doi: 10.1371/journal.pone.0115922) and reported that in contrast to the Ldlr-/-.Leiden mice, in WT mice the hypertriglyceridemia is lacking and with respect to liver histology, the WT mice develop a milder form of steatosis, but especially the hepatic inflammation is less prominent and fibrosis on a high fat diet (without supplemented cholesterol) is lacking.
Parts of these comparisons were already mentioned in the text of the discussion (paragraph discussing metabolic context, lines 445-450 in Track-changes version with All Markup) but in the subsequent paragraph discussing the histological phenotype of the liver this comparison with WT mice was indeed lacking and has now been added (lines 471-474). We thank you for pointing this out, since this comparison indeed strengthens our manuscript.
Response to previous OCA observations: We indeed only referred to our previous investigation of OCA in HFD fed Ldlr-/-.Leiden mice but now added a line to describe the most important finding as well (lines 524-526). In the previous study of OCA (10 mg/kg/d from week 24-34) in HFD fed Ldlr-/-.Leiden mice we found a tendency to reduce macrovesicular steatosis, while microvesicular steatosis remained unchanged. Furthermore, hepatic inflammation measured as number of inflammatory aggregates tended to be reduced and more refined quantitative immunohistochemical analysis revealed that this was significant for macrophages (F4/80 staining) and remained a tendency for T helper cells (CD4 staining) and cytotoxic T cells (CD8 staining). Hepatic fibrosis development was attenuated, but fibrosis was not reduced beyond the level of the start of intervention. So, in comparison to the HFD, the response of OCA on the FFD seems somewhat more pronounced on hepatic inflammation and fibrosis (although no head-to-head comparison of OCA in one experiment), which could be due to the higher induction of hepatic inflammation and fibrosis on the FFD. Furthermore, we did not observe a tendency on macrovesicular steatosis in the current study on FFD.
Response to point 1: Both the HFD and FFD increase liver weight as compared to the chow fed animals and we have now added liver weight (both absolute weight and relative to body weight) to Table 1.
Response to point 2: Hepatic inflammation was indeed evaluated in this study using the number of inflammatory aggregates that consists of mononuclear cells and polymorphonuclear cells. We previously analyzed these inflammatory aggregates in our model in more detail using specific stainings and found that these inflammatory aggregates stain positive for F4/80 (macrophages), CD4 (T helper cells) and CD8 (cytotoxic T cells) (see Morrison et al., Hepatol Commun. 2018 Dec; 2(12): 1513–1532. Published online 2018 Oct 29. doi: 10.1002/hep4.1270) and MPO (neutrophils) (see Morrison et al. Int J Obes (Lond). 2016 Sep; 40(9): 1416–1423. doi: 10.1038/ijo.2016.74). We have repeatedly identified these cell types in multiple unpublished in-house studies and studies for others. We found (using another animal model) that this mixed inflammatory cell type was typical for diet induced NASH (see also Liang et al., Lab Invest. 2014 May;94(5):491-502. doi: 10.1038/labinvest.2014.11) in contrast to NASH induced via HFD superimposed with LPS or IL-1β. Although indeed interesting, for this study we did not perform the different stainings, but there is no reason that the inflammatory aggregates differ from previous experiments which repeatedly confirmed the same aforementioned cell types. Indeed, consultation of a board-certified pathologist could identify monocytes and polymorphonuclear cells in the inflammatory aggregates essentially as reported previously. To follow up the suggestions of the reviewer, we did check the available transcriptome data for inflammatory markers, like CD11c, CD68, CD3, CD4, CD8, Ly6c, MCP-1 etc and for most markers a significant upregulation was found on the HFD and FFD vs. chow diet, with overall a slightly higher response for the FFD vs. HFD. In addition, we analyzed the transcriptome dataset for their cell-type specific gene expression profiles of leukocytes essentially as reported by Palmer et al., (BMC Genomics. 2006; 7: 115. Published online 2006 May 16. doi: 10.1186/1471-2164-7-115). These cell-type specific gene sets allows accurate assessment of presence of particular leukocytes in a tissue (thus constitutes a rapid alternative for multiple immunohistological stainings). We found higher -log(P values) for the B-cell population and T-cell population in HFD vs. chow and FFD vs. chow. Overall this supports the notion that the inflammatory aggregates consists of different inflammatory cells. In the revised manuscript we have added the remark that the inflammatory aggregates consists of monocytes and polymorphonuclear cells and refer to the manuscripts of Morrison et al., in which we analyzed this in more detail.
Response to point 3: OCA treatment did not significantly affect body weight (50.4 ± 2.6 g), food intake (14.0 ± 0.4 kCal/mouse/day) or liver weight (4.0 ± 0.5 g) as compared to the untreated FFD fed animals. We have now added this statement in the text (lines 304-306).
Response to point 4: This was an interesting suggestion, that we indeed were willing to follow up. Since we do not have a TUNEL staining running in our department we ordered a ready-to-use TUNEL assay. Unfortunately, the ready-to-use assay was not so ready to use as promised, since we were unable to get a proper staining. With the limited time we had for the revision of the manuscript, (we tried to optimize the staining) but we did not have enough time to get this staining running. So we could not add this. As an alternative, we have checked our transcriptomics data for apoptosis gene expression and found that most genes (73%) in apoptosis signaling were significantly upregulated in HFD vs. chow or FFD vs. chow, with a larger effect in the FFD than HFD. In addition, we have added the quantification of 4-hydroxy-2-nonenal (HNE) to the manuscript (revised figure 4d), a marker for oxidative stress that has been reported to induce apoptosis as well and found increased 4HNE levels with the HFD and FFD, all together these data suggest increased apoptosis on the diets vs. chow.
Reviewer 2 Report
I read with pleasure this interesting paper about Ldlr-/- mice as a possible pathogenic model for NASH. This paper can be divided into two parts. In the first part the Authors describe the animal model and its possible advantages over other animal models. In the second part the Authors use this model to assess the response to obeticholic acid (OCA) in the animal model.
While the first part is not entirely novel (other experiments have been performed in Ldlr-/- mice in the past), the second part is interesting and provide novel insights. The experiment is well described and with a good rationale. In particular, it is worth of note that the Authors also assess the role of liver fibrosis, which is the main driver of liver-related complications and death in humans. This is a significant strength of this paper in comparison with a large proportion of papers which report results of experiments in which the mice are sacrificed earlier and assess only the quantity of liver steatosis. The Authors should be also appreciated for testing OCA, which is the latest advance in this field and will be specifically licensed for the treatment of NASH in humans. This choice is another element of strength of the paper in comparison which other papers exploring other drugs which have not been specifically designed for NASH in humans.
The main weaknesses are: 1) a partial lack of novelty in the first part; 2) only male mica were tested while sexual hormones play an important role in NAFLD/NASH (the Authors can be partially justified as 99% of these experiments are conducted in the same way); 3) the model makes pathogenic experiments about the LDL pathway impossible, as the Authors state.
My comments regard only the last point: in particular, the Authors should speculate about which kind of drugs can not be tested in this model. This is not pure speculation, as some "old" drugs are being repurposed for the treatment of NASH and new trials in humans are expected. For example, will this model be suitable for comparing the effects of OCA with metformin, statins, PPAR-gamma inhibitors, GLP-1 agonists? This are useful information.
The conclusions are correctly drawn and deserve no specific comments.
Author Response
We kindly thank you for the time and effort it took to review the paper.
Response to point 1) and 2): Although we indeed have performed experiments with Ldlr-/-.Leiden mice before, this is the first study/manuscript in which we focus on the model itself and use a head-to-head comparison of two different high fat diets (which both emphasize different characteristics of the model ). In addition, we compare the model with a human NASH transcriptome signature and with the response to OCA.
With respect to the male mice being used, we completely agree that the general focus on one gender only is an issue that should be addressed. In this case, we chose the male mice because the male mice are more prone to develop obesity on high fat diets than female mice. Female Ldlr-/-.Leiden mice can and have been used for NASH studies as well though (on HFD they also develop NASH and hepatic fibrosis). For the information of the reviewer, in a collaboration with researchers in Spain, we have performed a study in female Ldlr-/-.Leiden mice on HFD and observed liver steatosis with inflammation and fibrosis, albeit somewhat less pronounced as in males. In juvenile obesity (using 6 w young mice), we found differences in fasting insulin (high plasma insulin levels in male mice only) and increased plasma lipids (higher triglycerides and cholesterol levels in males compared to females (Jacobs et al, Nutrients, 2019). The more outspoken metabolic syndrome phenotype is the reason why we used male mice for the present study which is not surprising because it is known that male mice with C57/BL6 background respond stronger to HFD compared to females.
Response to point 3: Several different nutritional and pharmaceutical interventions have been tested in our Ldlr-/-.Leiden mice before. Effective nutritional and nutraceutical interventions include protein hydrolysates and probiotics (Schoemakers et al., PLoS One, 2017); short-chain fatty acids such as butyrate and propionate (Arnoldussen et al., Int J Obes, 2017) or propionate (Tengeler and Gart et al., FASEB J, 2020); antioxidants including polyphenols and activators of mitochondrial function, L-carnitine and nicotinamide riboside (Luque-Sierra et al., Mol Nutr Food Res. 2018; Salic et al., Int J Mol Sci., 2019); consumer oils and marine phospholipid-rich oils (Gart et al., submitted; Alvarez-Amor et al., submitted) as well as lifestyle interventions (e.g. exercise; van den Hoek et al., poster EASL 2019 & paper in preparation). Pharmacological interventions include caspase-1 inhibitors (Morrison et al., Int J Obes, 2016), CXCR2/5 inhibitors (unpublished), inhibitors of apical sodium-dependent bile acid transporters (Salic et al., PLoS One, 2019), PPAR-activators such as lanifibranor (in preparation), pioglitazone (Radonjic et al., PlosOne, 2013) and rosiglitazone (Radonjic et al., PlosOne, 2013 & Mulder et al., Sci Rep., 2016), but also metformin, sitagliptin, glibenclamide, fenofibrate, atorvastatin, salicylate and rofexocib (all described in Radonjic et al., PlosOne, 2013). Furthermore, DGAT inhibitors (poster Salic et al., Keystone NASH symposium, 2019) and FXR activators such as OCA (Morrison et al. Hepatol Commun., 2018 & current manuscript). However, not all studies aimed to evaluate the effect on NASH/fibrosis. If we roughly divide the agents in development for NASH in different categories like, lipid modulators, metabolism modulators, incretins, PPAR modulators and anti-inflammatory or anti-fibrotic agents, we would expect that for most of these categories our Ldlr-/-.Leiden mice could be useful, except for lipid modulators that rely on an intact LDL receptor for their mode of action. For the latter category the lack of LDL receptor could be a problem for some agents like statins that require an intact LDL receptor pathway and these agents could better not be tested in the Ldlr-/-.Leiden model. Of note, statins have been tested in Ldlr-deficient mice in the past to investigate pleiotropic effects of these compounds which illustrates that a general statement is difficult because suitability of the model depends on the research question.
Without going into too much detail and discussing all interventions tested in the model, we have now added lines in the manuscript describing the categories of interventions we expect can be evaluated and added the statement that for lipid modulating interventions like statins the model is not useful (lines 575-579 in Track changes version with All Markup).
Reviewer 3 Report
In this manuscript, the authors investigated the pathological effects of high fat diet (HFD) and fast food diet (FFD) in Ldlr-/-.Leiden mice. The mice on these diets developed pathological conditions similar to those observed in human NASH. Animal models reflecting human pathology like this model mouse are quite valuable and useful for the study of NASH. Overall, this study is interesting, and would be benefit for clinical researchers. However, the scope of this journal is focused on experimental cytology, including cell biology and physiology, molecular biology, and biophysics rather than on clinical and epidemiological studies. Therefore, the authors should substantially revise your study to publish in Cells. In addition, several issues remain to be addressed as follows:
Major:
- Why did the authors use two different diets, and decide the composition? Detailed explanation on the diets used should be provided.
- Please discuss why blood glucose levels were not changed by HFD and FFD. From Table S1, it seems that Ldlr-/-.Leiden mice already have weak hyperglycemia. If so, Ldlr-/-.Leiden mice fed on HFD or FFD had better be compared with wild-type mice.
- To rule out the possibility that the mild and transient increase in plasma insulin levels is due to reduced diet consumption, the authors should record and report it.
- Hepatic inflammation also showed transient increase. OCA treatment should be performed from 12 to 22 weeks.
- The authors well discussed the difference in insulin resistance between the diets from pathophysiological viewpoints. However, the readers of "Cells" are more interested in molecular mechanisms. In addition, the downregulation of complex V genes was observed after the development of insulin resistance. Thus, it is necessary to verify at a molecular level whether complex V contributes the development of insulin resistance in this mouse model.
- Similarity of gene expression profiles to human NASH is great advantage as an animal model. However, to gain mechanistic insights into this NASH model, the authors should perform RNA-seq in a time course manner. For instance, the mechanism by which HFD induced stronger inflammatory response in adipose tissue, while FFD did that in liver, would be useful to establish more precise animal model, and to deepen our understanding NASH. At least, mice on FFD diet should be analyzed before 28 weeks where hepatic inflammation and hyperinsulinemia are slightly ameliorated.
Minor:
- lines 308. n=247 –> 237
- Figure 4(b). What do red, yellow and gray stars indicate?
Author Response
We kindly thank you for the time and effort it took to review the paper.
Response to point 1: For our model we made the decision to use translational obesogenic diets to mimic the diets to which many subjects (including NASH patients) in modern societies are exposed to. Therefore we excluded diets deficient in essential nutrients or dietary components (like e.g. methionine-choline deficient diets) or diets that were supplemented with supraphysiological amounts of cholesterol. We previously tested several different diets (see also Abe et al., Biol Open. 2019 May 1;8(5):bio041251. doi: 10.1242/bio.041251.) including high fat and high fructose diets and found the diets used in the current study to be most translational, in respect to mimicking the macronutrient composition of human diets as well as in respect to inducing NASH with fibrosis in a Metabolic syndrome-like context. For instance, the fat content of an experimental diet should not exceed 25% w/w equaling about 45% energy from fat, which is reached in diets consumed in Crete or Finland (see also Hu FB, Manson JE, Willett WC. Types of dietary fat and risk of coronary heart disease: a critical review. J Am Coll Nutr 2001;20:5–19. DOI: 10.1080/07315724.2001.10719008). In the present study we used diets that differed regarding the source of the dietary fat (lard or milk fat) and the carbohydrate source (primarily sucrose or fructose).
Since our study shows that the different diets emphasize different characteristics of the model (insulin resistance with HFD, hepatic fibrosis with FFD), it’s possible that even different diets, or slightly different diet compositions of the current diets, may affect the NASH profile of the model even further. However, since there are endless possibilities to vary in diets/diet composition, the decision for the current diets was a pragmatic decision based on literature and our previous experience with different other diets in our model and other models. Diets with high (supraphysiological) cholesterol content were avoided because of recent criticisms regarding the bias to cholesterol-mediated processes and cholesterol-dependent inflammatory processes. Also diets with extreme fat content (e.g. 40% w/w equaling >60En%) or extreme protein content should in our opinion be avoided because these diets introduce metabolic adaptations and biases that hamper translation to the human situation. In the revised introduction we have added a sentence that provides the rationale of our dietary choice. We thank the reviewer for raising this point.
Response to point 2: Our model on the high fat diets (within the time frame investigated) does not progress to a real diabetic model (i.e. loss of pancreatic insulin production), but rather a very pronounced insulin resistant state with, as compared to the chow diet, strongly increased plasma insulin levels but similar glucose levels. It is important to realize that this study shows that insulin levels can increase to >10 ng/mL on HFD indicating very pronounced hyperinsulinemia in these mice (from longitudinal studies we know that hyperinsulinemia persists up to 35 weeks throughout HFD feeding). The reviewer stated that Ldlr-/-.Leiden mice have already weak hyperglycemia on chow diet and that they better could be compared to wild-type mice of the same C57BL/6 background. We fed wildtype C57BL/6 mice the same chow and HFD diet as in the present study (Mulder et al, PLoS One, 2016; Figure 2C) and observed comparable fasting glucose levels on chow of about 8 mM. This indicated that the chow diet itself may result in metabolic adaptations in mice that suggest mild hyperglycemia. We have therefore also tested different kind of ‘healthy’ reference diets (chow vs. D12450H, D12451K and AIN93G, all with different carbohydrate compositions and all proposed to have a normal ‘healthy’ composition) in our Ldlr-/-.Leiden model. Indeed with the current chow diet blood glucose levels are slightly higher as compared to other healthy reference diets. As compared to some of the other ‘healthy’ reference diets, the Ldlr-/-.Leiden mice on HFD or FFD do have increased blood glucose levels. Nevertheless, despite the indeed slightly elevated glucose levels on the chow diet used herein, we continued our studies with the chow diet, since on this diet we did not observe an increased gut permeability, increased plasma cholesterol/triglycerides or NASH phenotype in a few mouse per group (increased steatosis, hepatic inflammation), that was observed for some of the other reference diets. In other words, diets that are considered to be ‘healthy reference diets’ do have an effect in Ldlr-/-.Leiden mice and can counterintuitively induce hypercholesterolemia or gut permeability or NAFLD/NASH phenotype. As many researchers are not aware of the effect of reference diet choices, a comprehensive analysis of the effects of the commonly used reference diets is a topic that we will follow up in the future. Of note, the effects of such reference diets will differ for different animal models (e.g. wt C57BL/6 vs. ob/ob vs. KKAy as reported by Abe et al., Biology Open, 2019) which is why it is difficult to draw general conclusions on reference diets.
Response to point 3: Diet consumption was measured three times during the study (in 3 or 4 cages per group) and was similar. We have added the average food intake in Table 1 now.
Response to point 4: Hepatic inflammation indeed seems to be a dynamic process and advanced stages of NASH can have less inflammation then earlier stages (see also review of Koyama et al., J Clin Invest. 2017;127(1):55–64. doi:10.1172/JCI88881.) In hindsight and with respect to hepatic inflammation OCA treatment during week 12-22 would indeed be very interesting! However, OCA treatment is not confined to an effect on inflammation only, but also affects metabolism and fibrosis, which all have their own dynamic pattern during the high fat diet induced NASH development in our model. Hence, an intervention period that would be optimal to inhibit the development of lobular inflammation (12-22 wk) is less optimal to assess OCA effects on developing fibrosis and vice versa. In the present study we chose fibrosis because of its clinical relevance as endpoint. Also in NASH patients, timing of treatment with OCA during the peak of hepatic inflammation cannot be performed. As we do not have time points between t=12 and t=22 we also do not know exactly when the peak of hepatic inflammation occurred in the FFD-fed Ldlr-/-.Leiden model. We are planning to perform a separate longitudinal study with multiple time points to assess the exact dynamics of inflammation and fibrosis accurately. Our observation that more inflammatory aggregates were found at 18 wk of FFD feeding compared to 28 wk supports the view that hepatic inflammation is dynamical and that advanced stages of NASH may have less inflammation than earlier stages as reported by Koyama and Brenner (review JCI 2017; DOI: oi:10.1172/JCI88881) and we have mentioned this in the revised discussion. Despite of our limited knowledge of the exact dynamics of inflammation and fibrosis, the tested OCA treatment (from week 18-28 ) significantly lowered hepatic inflammation and fibrosis in our model.
Response to point 5: It is important to mention that the insulin resistance we discuss in our manuscript is not specifically hepatic insulin resistance, but rather an overall, whole-body insulin resistance based on increased plasma insulin levels and blood glucose levels (or HOMA-IR index). For determination of hepatic insulin resistance a hyperinsulinemic euglycemic clamp should have been performed, essentially as we did in earlier studies in the context of diet-induced obesity (Kleemann et al., PLOS one 2010) and NAFLD (Mulder et al., PLOS one 2017). Although mitochondrial dysfunction has been implicated in the development of insulin resistance, we only show mitochondrial dysfunction based on gene expression levels and at the end of the experiment. We indeed cannot determine whether the mitochondrial dysfunction preceded the (hepatic) insulin resistance and even if we do now the sequence of these developments in time, it remains difficult to demonstrate a causal contribution of the mitochondrial dysfunction to insulin resistance. As mentioned in our discussion we do think the difference in fat distribution between the two diets plays a role in the difference in insulin resistance. We previously showed the causal role of inflamed perigonadal WAT to NASH development by demonstrating that via surgically removal of inflamed perigonadal WAT, the hepatic inflammation component of NASH was attenuated (see Mulder et al., Int J Obes (Lond), 2016. 40(4): p. 675-84). WAT and especially visceral/mesenteric WAT has been associated with insulin resistance and therefore we think the different fat distribution pattern between the two diets plays a role and leads to a mechanistic slightly different balance in the induction of NASH in the model. With a larger role for WAT and insulin resistance with the HFD. For the FFD we observed on gene expression level a down-regulation in mitochondrial complex V. As mitochondrial dysfunction can lead to oxidative stress and inflammation, we think the mitochondrial dysfunction plays a larger role with the FFD in the induction of hepatic inflammation and fibrosis. As we agreed with the reviewer that for the readers of “Cells” it would be important that we not only show the mitochondrial dysfunction on a gene expression level, but also on a more molecular level, we have now performed additional measurements and measured 4-hydroxynonenal (HNE), which is a marker for oxidative stress that can trigger pro-inflammatory cytokines. Quantification of a 4 HNE staining in the liver has been included in the revised Result section and revealed that this was indeed significantly higher for both high fat diets as compared to chow and tended to be more increased with the FFD. These results are therefore in line with the gene expression results and have now been added to the manuscript.
Response to point 6: We agree that RNA-seq in a time course manner can add useful information for the NASH development and add to the mechanistic insight of the model. For the HFD, we previously analyzed this (see also van Koppen at al., Cell Mol Gastroenterol Hepatol, 2018. 5(1): p. 83-98 e10) and analyzed the temporal dynamics of the key processes involved in NASH development and we found first processes of lipid metabolism being activated (from week 6 onward), while the inflammatory, oxidative stress, and fibrotic response were activated from week 12 onward. Interestingly, fibrosis pathways were found to be activated before actual fibrosis/collagen deposition in the liver was observed. For FFD we have not performed a time course that included transcriptomics yet, and we agree that this would be interesting to do so. However, this was not possible within the time frame of the revision.
In the current study with both diets transcriptomics being analyzed at t=28 weeks we found remarkable similarity on pathway level between both diets, while with respect to NASH pathology in the liver the FFD had much more pronounced hepatic inflammation and fibrosis as compared to the HFD. As mentioned above, we think there are slight mechanistic differences in the development of NASH between the two diets, or more precise, difference in the balance/role of different processes that induce NASH. We think the different distribution of fat across WAT depots may play a role, because FFD resulted in more perigonadal WAT which has been shown to causally affect inflammation in liver (Mulder et al., Int J Obes,, 2016). Whether or not HFD induced a stronger inflammatory response in adipose tissue remains debatable, since the number of inflammatory aggregates/mm2 pgWAT was indeed increased in HFD, but the total absolute weight of pgWAT was much higher in FFD thus resulting in a similar inflammation when taking the whole adipose depot into account. Furthermore, it remains difficult to determine the exact differences in NASH induction of the two diets via a time-course transcriptomic approach of the liver, since the differences in adipose tissue distribution and insulin resistance between both diets might not be revealed by hepatic transcriptome analysis. For instance, insulin resistance pathways are known to be predominantly regulated on a protein level via phosphorylation/dephosphorylation changes and kinase activity.
With respect to the mitochondrial dysfunction, that we think, plays a more important role in the NASH induction with the FFD, we were able to demonstrate the difference between the diets by using end-point transcriptomics only, and the obtained results are consistent with the immunohistochemically quantified levels of 4-HNE as outlined above. It would however indeed be very interesting to perform a time-course study with FFD and preferably with transcriptomics being performed in multiple metabolic active organs, like liver, WAT and muscle. This was obviously beyond the scope of the current manuscript and might be a good follow-up suggestion. We do intend to perform a more detailed time-course study on the FFD as mentioned above in analogy to the study of van Koppen at al., Cell Mol Gastroenterol Hepatol, 2018. 5(1): p. 83-98 e10.
Response to minor point 1: Thank you for noticing this mistake, this has now been corrected.
Response to minor point 2: Unfortunately the stars that should have been above the bars to indicate to which of these categories they belong were not copied. This has now been changed. Thank you for noticing this mistake!
Round 2
Reviewer 1 Report
The authors responded to the reviews satisfactorily.
Reviewer 2 Report
The authors provided a comprehensive and constructive reply to the point I previously raised. Their response to point 3 will surely help the readers to understand the correct clinical setting of this study.
I have no further comments.
Reviewer 3 Report
The authors have adequately addressed my comments. I have no further comments.